# PEAR, a flexible fluorescent reporter for the identification and enrichment of successfully prime edited cells

Dorottya Anna Simon[1,2,3†], András Tálas[1*†], Péter István Kulcsár[1,4], Zsuzsanna Biczók[1,3], Sarah Laura Krausz[1,3], György Várady[1], Ervin Welker[1,5*]

[1]Institute of Enzymology, Research Centre for Natural Sciences, Budapest, Hungary; [2]ProteoScientia, Budapest, Hungary; [3]School of Ph.D. Studies, Semmelweis University, Budapest, Hungary; [4]Biospiral-2006, Szeged, Hungary; [5]Institute of Biochemistry, Biological Research Centre, Szeged, Hungary

**Abstract** Prime editing is a recently developed CRISPR/Cas9 based gene engineering tool that allows the introduction of short insertions, deletions, and substitutions into the genome. However, the efficiency of prime editing, which typically achieves editing rates of around 10%–30%, has not matched its versatility. Here, we introduce the prime editor activity reporter (PEAR), a sensitive fluorescent tool for identifying single cells with prime editing activity. PEAR has no background fluorescence and specifically indicates prime editing events. Its design provides apparently unlimited flexibility for sequence variation along the entire length of the spacer sequence, making it uniquely suited for systematic investigation of sequence features that influence prime editing activity. The use of PEAR as an enrichment marker for prime editing can increase the edited population by up to 84%, thus significantly improving the applicability of prime editing for basic research and biotechnological applications.

**\*For correspondence:**
talas.andras@ttk.hu (AT);
welker.ervin@ttk.hu (EW)

[†]These authors contributed equally to this work

**Competing interest:** The authors declare that no competing interests exist.

## Editor's evaluation

Prime editing (PE) is an emerging precision genome editing technology. It is based on the fusion of reverse transcriptase and Cas9 nickase guided by a modified guide RNA (pegRNA) to select the target site in the genomic DNA. The design of pegRNAs is not trivial and PE's editing efficiencies are low and highly dependent on the context. The authors of this manuscript have developed a surrogate reporter-based approach (PEAR) not only to identify but also to enrich for the cells that have been edited by PE. This approach will provide the means to better understand the molecular basis of PE (which could lead to the development of new, improved PEs) and improve the efficiency of PE mediated genome editing. This study, describing PEAR is of interest to the researchers in the basic biological, biomedical and agricultural sciences fields.

## Introduction

Discovery of CRISPR systems in bacteria and archaea has not only drastically increased the spectrum of organisms that we can genetically modify but it has also equipped us with a simple tool for the introduction of a variety of different types of modifications into the genome (*Cong et al., 2013*; *Mali et al., 2013*; *Wang et al., 2013*). Many of the CRISPR-based approaches rely on generating double-stranded DNA breaks that are usually accompanied by genome-wide off-target editing (*Tsai et al., 2015*; *Kim et al., 2015*; *Kleinstiver et al., 2016b*; *Tsai et al., 2017*). Recently, more precise CRISPR tools—base (*Komor et al., 2016*; *Gaudelli et al., 2017*) and prime (*Anzalone et al.,*

*2019*) editors—have been developed, and these enzymes can introduce modifications into the DNA with a base pair resolution without the requirement of donor DNA templates or the introduction of double-stranded DNA breaks. While current base editor (BE) variants provide efficient editing, they are restricted to certain substitution mutations (*Komor et al., 2016*; *Gaudelli et al., 2017*; *Kurt et al., 2021*; *Zhao et al., 2021*).

In contrast, prime editors (PEs) can introduce all types of substitutions and/or precise insertions/deletions (indels), however, their efficiency lags behind that of BEs. Prime editor 2 (PE2) consists of a nickase version of *Streptococcus pyogenes* Cas9 (SpCas9) fused to a reverse transcriptase enzyme (*Anzalone et al., 2019*). The fused reverse transcriptase can extend the nicked DNA strand using an RNA template that is located on the 3′ terminus of an extended single-guide RNA (sgRNA), called the prime editing sgRNA (pegRNA). By careful design of the pegRNA, modifications can be introduced downstream of the nick generated on the non-targeted DNA strand. The 3′ end of the pegRNA, that is complementary to the non-targeted DNA strand, consists of the reverse transcriptase template (RT) and the primer binding site (PBS). The RT, which contains the mutation(s) to be introduced, also comprises the protospacer adjacent motif (PAM). The PBS is complementary to the 3′ end of the nicked non-targeted DNA strand, with which it forms a hybrid DNA-RNA helix. The optimal length of the PBS and the RT varies from target to target and requires extensive optimization for each different target. The efficiency of prime editing can be increased by nicking the non-edited strand (prime editor 3—PE3) at the expense of potentially generating unwanted indels at the targeted locus (*Anzalone et al., 2019*). The effect of the nicking varies depending on its position and it requires further optimization. In some cases, the efficiency of prime editing can be further increased by also altering the PAM sequence with the desired mutation this may prevent the PE from nicking the sequence again. This method can reduce the chance of introducing unwanted indels alongside the intended mutations, thus increasing the occurrence of precise modifications (*Anzalone et al., 2019*).

The development of improved PE variants can be forecasted in view of the history of BEs which are now available with highly improved features and efficiency (*Kim et al., 2017*; *Komor et al., 2017*; *Koblan et al., 2018*; *Huang et al., 2019*; *Gehrke et al., 2018*; *Zafra et al., 2018*; *Gaudelli et al., 2020*). The aim of our study was to develop a reporter system for an easy, fluorescence-based detection of prime editing outcomes that allows maximum flexibility for the target sequences and pegRNA designs to be tested on it. Several systems have been developed for reporting base editing that use BEs for the generation or alteration of a start or stop codon (*Katti et al., 2020*; *Wang et al., 2020*), or to rescue a disruptive amino acid and subsequently recover functions of antibiotic resistance genes or fluorescence proteins (*Martin et al., 2019*). Alternatively, a non-synonymous mutation in the chromophore of a fluorescent protein that induces fluorescence spectral change has also been explored as a method to monitor base editing activity (*Coelho et al., 2018*; *Standage-Beier et al., 2019*). Reporter systems have also been demonstrated for prime editing, but they were restricted to very few target sequences (*Lin et al., 2020*; *Sürün et al., 2020*) and/or showed a rather low signal (*Katti et al., 2020*).

We also wanted to find out whether prime edited mammalian cells could be identified and enriched by using a plasmid-based surrogate marker for chromosomal DNA modifications, as demonstrated before with Cas nucleases (*Ramakrishna et al., 2014*; *Grav et al., 2015*) and BEs (*Katti et al., 2020*; *Coelho et al., 2018*; *St Martin et al., 2018*; *Tálas et al., 2021*).

## Results

We aimed to develop a reporter system that possesses several key features. It should be a transient plasmid-based system that is not restricted to one or a few cell lines nor does it require extensive work, such as the generation of cell lines. It needs to be based on a gain-of-function fluorescent signal with minimal background so it can be detected in the timeframe of a transient system. The sequence requirements of efficient prime editing are not yet fully understood (*Anzalone et al., 2019*; *Kim et al., 2021*); thus, for the widespread application of a reporter system, it is crucial that the sequence of the target and the flanking nucleotides of the position to be edited can be freely interchanged. We have recently developed a base editor activity reporter (BEAR), which meets these criteria (*Tálas et al., 2021*), and it has the potential to be converted into a tool suitable for reporting on prime editing activity. BEAR is based on a split GFP protein separated by the last intron of the mouse *Vim* gene. The sequence of the functional splice donor site (or 5′ splice site) is altered in such way, that splicing and therefore the GFP fluorescence is disrupted, however, they can be restored by applying adenine or

cytosine BEs (*Figure 1—figure supplement 1A*). Sequence alterations in the intronic sequence are well tolerated, which gave us the flexibility to investigate numerous target sequences. Using the information acquired during the development of the BEAR system (*Tálas et al., 2021*), we have designed a plasmid that contains an inactive splice site, which can be activated by the action of a PE harboring an appropriately designed pegRNA (*Figure 1A*). Cells containing plasmids with activated splice site sequences will then be able to efficiently express GFP, which can be quantified by flow cytometry. Therefore, we name assays that exploit this design as prime editor activity reporters (PEARs).

There are differences between the features of BEAR and PEAR. The editing windows of BEs exploited for the BEAR system are located PAM distal, 5′ of the nick (*Figure 1—figure supplement 1B*). PEs on the other hand can introduce mutations 3′ of the nick, a region that contains the PAM sequence (*Figure 1A*, *Figure 1—figure supplement 1C*). To ensure maximum flexibility for the PE target, we kept the target sequence in the intron by placing the PAM to the complementary strand. As a result, the target sequence can be freely adjusted in its entire length (*Figure 1A*). Thus, while BEAR could be used with a few million of different target sequences, PEAR offers unrestricted sequence variations for the entire SpCas9 target. On the contrary, PEAR is similar to BEAR in that it is permissive with respect to the interchangeability of the targeted nucleotides (here the splice site and the splice site flanking nucleotides; *Tálas et al., 2021*). No other fluorescence-based reporter systems have demonstrated such degree of flexibility before (*Katti et al., 2020*; *Lin et al., 2020*; *Sürün et al., 2020*).

To identify the optimal PBS, RT, and complementary nick combination, HEK293T cells were transfected with the target plasmid (named PEAR-GFP) carrying the inactive splice site sequence to be edited, and several combinations of plasmids coding various pegRNAs, sgRNAs (targeting the PEAR-GFP plasmid), and PE2 protein. The number of GFP-positive cells was then measured by flow cytometry (*Figure 1—figure supplement 1D*). The heatmap in *Figure 1B* shows that the PEAR-GFP plasmid can be efficiently edited by many of these combinations of PBS, RT, and nicking position. The most efficient editing occurred, when a 10-nucleotide long PBS (PBS-10) and a 24-nucleotide long RT region (RT-24) were applied. *Figure 1B* also shows that in line with expectations, editing efficiency is generally higher when the complementary strand is nicked, than when it is not (*Anzalone et al., 2019*). The most effective complementary strand nicking site was position +17 in the case of most PBS and RT length combinations. In all conditions, PBS-10 gave greater values than PBS-13 and PBS-16. Comparing the RTs, the efficiency of the editing was in the order of RT-24>RT-33>RT-16 in most conditions (16 out of 18). These results are consistent with the concept that the effect of the length of PBS and RT on editing efficiency is primarily independent of one another (*Kim et al., 2020*). It is also in line with the current optimization practice, where first, the length of the PBS is optimized using a given RT length, and then the length of the RT is optimized using the PBS selected in the first step. Thus, these experiments support the hypothesis, that PE efficiency is governed by the same factors when using a PEAR plasmid, as demonstrated earlier on chromosomal targets (*Anzalone et al., 2019*; *Kim et al., 2021*).

Before proceeding to the further characterization of the sequence dependence of PEAR, we examined whether the fluorescent signal originates exclusively from the recovered splice site sequence. PEs themselves generate various amounts of indels that may lead to the incidental restoration of fluorescence. In this experiment, to report on prime editing, we used a previously constructed plasmid (*Tálas et al., 2021*) that expresses a split mScarlet (BEAR-mScarlet). We wanted to simultaneously detect caused indels in a cell line by the means of a disruption assay, as well as prime editing on the BEAR-mScarlet plasmid. We transfected the BEAR-mScarlet plasmid along with either an SpCas9 nuclease, a nickase or a PE protein-coding plasmid into HEK293.EGFP cells (harboring the coding sequence of EGFP integrated into the genome). We also transfected a pegRNA to correct the splice site on the BEAR-mScarlet plasmid along with sgRNA(s) targeting the GFP coding sequence. This design allowed us to co-monitor indel generation via GFP disruption and prime editing via restoring mScarlet fluorescence on the BEAR-mScarlet plasmid. As expected, while all three variants—SpCas9 nuclease, nickase, or PE—demonstrated GFP disruption activity, only PE restored mScarlet fluorescence, indicating that PEAR specifically reports on prime editing activity and is insensitive to the potential presence of indel background (*Figure 1—figure supplement 1E*).

Using the optimal PBS-RT combination found in *Figure 1* constructs with different inactive splice site sequences were edited by PE2 to the same active splice site sequence (as in *Figure 1B*) by either

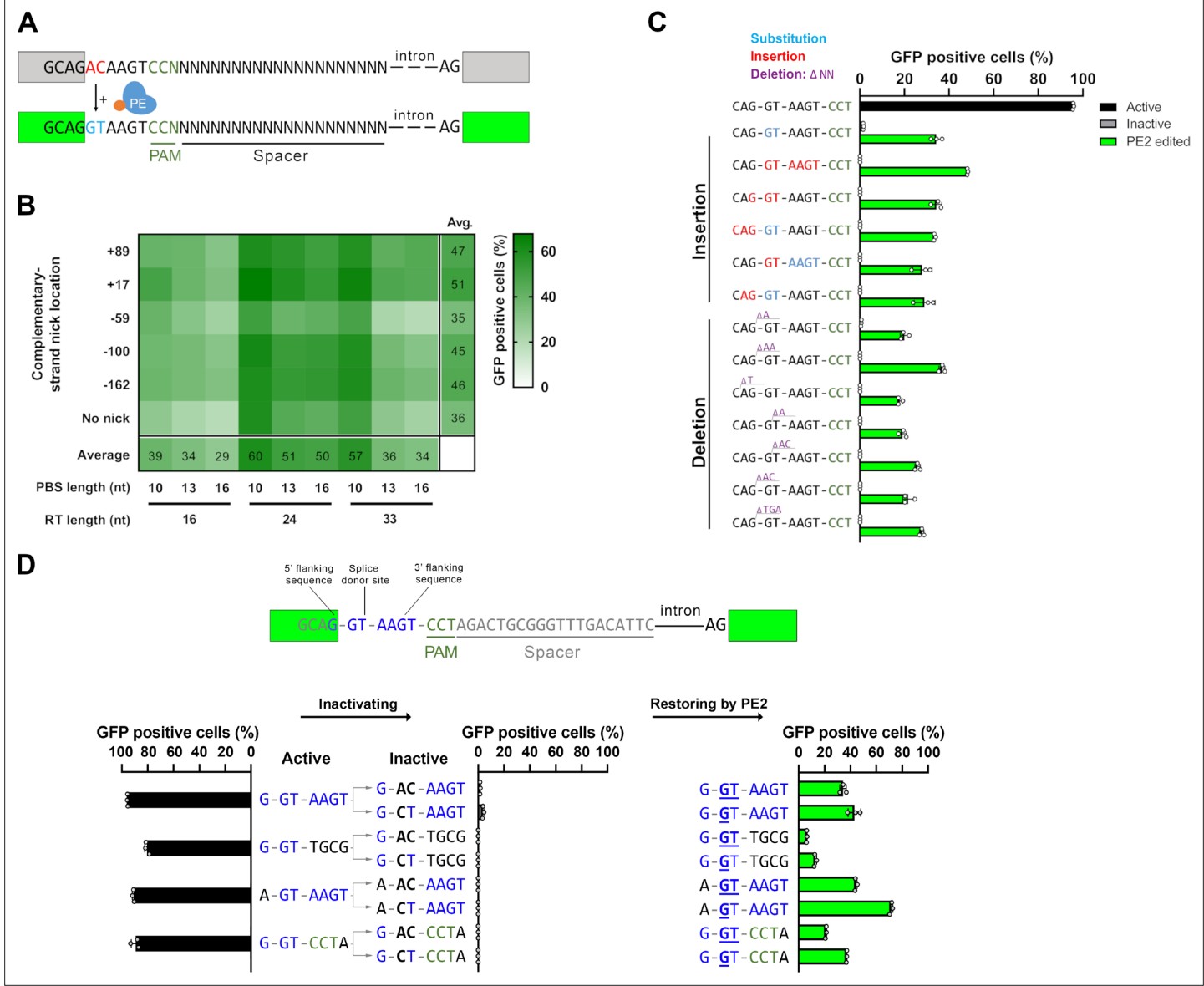

**Figure 1.** Principle of the prime editor activity reporter (PEAR) assay. (**A**) Schematic of the PEAR. The mechanism of PEAR is based on the same concept as BEAR (see *Figure 1—figure supplement 1A and B*), and it contains the same inactive splice site, shown in (**A**). PE can revert the 'G-AC-AAGT' sequence to the canonical 'G-GT-AAGT' splice site. Prime editing occurs downstream of the cut site in the target; hence, this method enables us to position the spacer sequence within the intron, thus, the entire length of the spacer (shown as 'N'-s) is freely adjustable in PEAR. The altered bases of the splice site are shown in red, the edited bases are shown in blue, and the PAM sequence is shown as dark green letters, the nCas9 is blue and the fused reverse transcriptase is orange. (**B**) Optimization of PBS, RT, and complementary DNA strand nicking on the PEAR-GFP plasmid. The heatmap shows the average percentage of GFP-positive cells of three replicates of transfections with the PEAR-GFP plasmid in combination with PE vectors which also contain the different pegRNAs and the sgRNAs for secondary nicking. The position of the second nick is given in relation to the first nick. Positive values indicate 3', negative values indicate 5' direction on the targeted DNA. When no second nick was introduced, it is indicated as 'no nick'. (**C, D**) Flow cytometry measurements of HEK293T cells transfected with active (positive control) or with inactive PEAR plasmids either along with nCas9 for negative controls or with PE2 vectors (black, gray, or green columns, respectively). (**C**) In the PEAR system, GFP fluorescence can be restored from various inactive splice sequences not only by substitutions but also by insertions/deletions. Each inactive sequence (gray columns) was corrected (green columns) to the canonical splice site by PE2 with the optimal pegRNA (pegRNA 1) from (**B**). The edited sequences are shown next to the columns. Red indicates inserted, blue substituted, and purple deleted bases. (**D**) Prime editing can result in various active splice sequences, further demonstrating the sequence flexibility of the PEAR system. Based on *Tálas et al., 2021*, four additional active splice site variants were selected. To generate suitable inactive plasmids, splicing was disrupted by systematically replacing the 5'-GT splice donor site to 5'-AC and 5'-CT (negative controls with inactive sequences, gray columns). With the appropriate pegRNAs, PE2 was able to restore GFP fluorescence in every case (green columns). Letters highlighted in blue indicate the bases of the canonical splice donor site and the flanking sequences which influence splicing the most: 5'-G-GT-AAGT-3'; the altered

*Figure 1 continued on next page*

*Figure 1 continued*

bases of the inactive sequences are bold, and bases in the active sequences that are reverted by PE2 are underlined. (**C, D**) Columns represent means ± SD of three parallel transfections (white circles). For all measured values see *Figure 1—source data 1*.

The online version of this article includes the following source data and figure supplement(s) for figure 1:

**Source data 1.** Measured values for *Figure 1*.

**Figure supplement 1.** The differences between the base editor activity reporter (BEAR) and PEAR assays.

**Figure supplement 1—source data 1.** Measured values for *Figure 1—figure supplement 1*.

deleting or inserting nucleotides in six and seven cases, respectively (*Figure 1C*). Interestingly, there seems to be no connection between the efficiency and the complexity of the editing. *Figure 1C* demonstrates that all types of sequence alterations can be examined using the PEAR system.

The versatility of the PEAR assay, with respect to sequence modifications, lies not only in the fact that an active splice site can be restored from seemingly any inactive sequence as long as the change can be efficiently implemented by prime editing, but also that the active splice site sequence itself can be varied. To demonstrate this, we corrected three additional active splice sites from different inactive sequences (*Figure 1D*).

In order to verify that the same factors influence prime editing on a plasmid as on genomic targets, we compared prime editing efficiency on previously constructed plasmids and two HEK293T cell lines (HEK-BEAR-GFP and HEK-BEAR-mScarlet—*Figure 2—figure supplement 1*; *Tálas et al., 2021*) that contain a chromosomal copy of these plasmids harboring an interrupted GFP or mScarlet sequence, respectively. As the plasmids (BEAR-GFP and BEAR-mScarlet) contain the same sequence around the inactive splice site as in the cell lines, they can be compared pairwise. The sequences to be edited in the two cell lines were designed to be used with base editing that requires a PAM sequence 5′ upstream of the edit. By contrast, prime editing requires a PAM position 3′ downstream. Thus, we needed to identify another PAM site that would allow the application of prime editing on these sequences. For BEAR-GFP, there is just one PAM, but for BEAR-mScarlet there are two PAMs in the primary editing window, and this gave the opportunity to perform the comparison for two targets for BEAR-mScarlet.

In case of the GFP expressing cells, we tested 36 different conditions for both the cell line and the BEAR-GFP plasmid: all possible PBS-RT-second nick site combinations. The heatmap in *Figure 2A* shows that PBS-13 and RT-22 or RT-26 in combination with the second nick site at position +103 results in maximum efficiency for the BEAR-GFP plasmid. In the absence of a second nick, the efficiency of editing was considerably lower. The most ideal conditions for prime editing were the same both in the plasmid and in the cell line (*Figure 2A*). The overall pattern of the two heatmaps is indisputably similar, exhibiting strong correlation (r=0.89, *Figure 2—figure supplement 2A*). In case of the cell line, a somewhat lower general editing efficiency is apparent compared to that with the plasmid. This difference is likely due to the much higher copy number of the plasmids present in the cell.

In case of the mScarlet expressing cells, we compared the efficiency of prime editing with both target 1 (*Figure 2B*) and target 2 (*Figure 2—figure supplement 2B*), by exploring 36 different combinations of PBS, RT, and second nick site for each target. Target 1 gave higher prime editing efficiencies, with the top working PBS being 10 nucleotides long in the case of both the BEAR-mScarlet plasmid and the cell line (*Figure 2B*). The two heatmaps exhibit a remarkably similar pattern, and there is a strong correlation of the editing efficiency in the cell line and the BEAR plasmid (r=0.93; *Figure 2—figure supplement 2C*). Similar outcomes can be seen with target 2 where the best PBS length is also 10 nucleotides long for both the BEAR-mScarlet plasmid and the cell line (*Figure 2—figure supplement 2B*). Comparing plasmid and genomic results also revealed a remarkably similar pattern (r=0.92; *Figure 2—figure supplement 2D*). With the same intron being in all three plasmids, we used the same three sgRNAs for secondary nicking in combination with all pegRNAs and plasmids or cell lines. For all three targets studied, regardless of whether in plasmids or in cell lines, the three secondary nick positions affected the effectiveness of prime editing in the same way. Collectively, these results strongly imply that the main features of prime editing and factors affecting its efficiency are reflected accurately within our system.

Post-transfection selection of edited cells by fluorescence-activated cell sorting (FACS) or antibiotics have proven to be effective methods for the enrichment of genetically modified cells within a

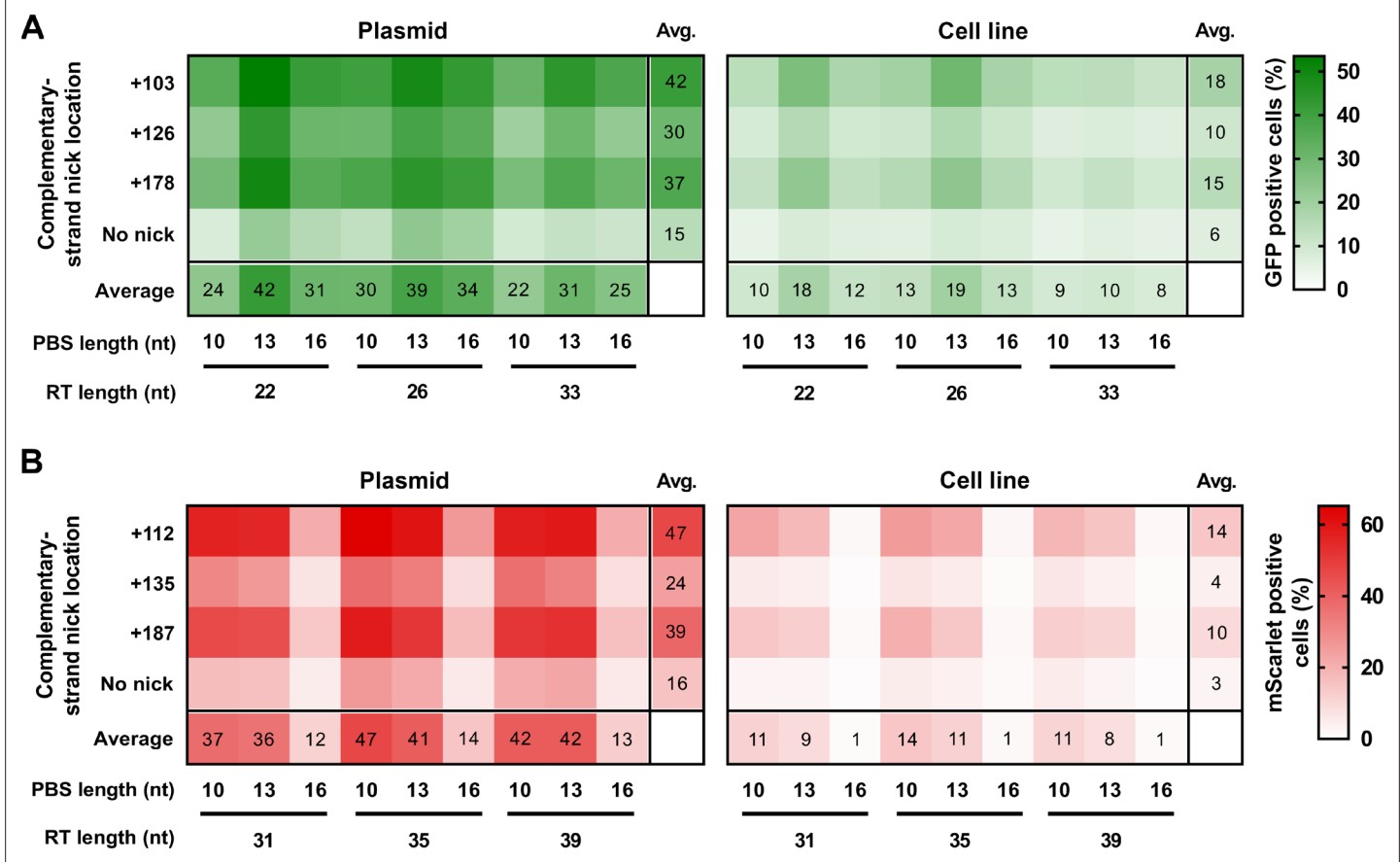

**Figure 2.** Prime editing with PEAR in a genomic and plasmid context. PEs targeting BEAR sequences either located in plasmids (Plasmid) or incorporated into the genome (Cell line) were transfected into cells alongside different sgRNAs for different complementary strand nick locations. The heatmaps show the average percentage of GFP (**A**) and mScarlet (**B**) positive cells derived from three replicates. For all measured values see *Figure 2—source data 1*. BEAR, base editor activity reporter; PEAR, prime editor activity reporter.

The online version of this article includes the following source data and figure supplement(s) for figure 2:

**Source data 1.** Measured values for *Figure 2*.

**Figure supplement 1.** Gating examples from HEK-BEAR cell lines.

**Figure supplement 2.** Prime editing of the BEAR-mScarlet plasmid with target 2.

**Figure supplement 2—source data 1.** Measured values for *Figure 2—figure supplement 2*.

population, given that the SpCas9 coding plasmid co-expressed a protein that is fluorescent or bears antibiotic resistance (*Katti et al., 2020*; *Standage-Beier et al., 2019*; *Ramakrishna et al., 2014*; *Grav et al., 2015*; *St Martin et al., 2018*; *Tálas et al., 2021*). Experiments have also shown that selecting for cells with a surrogate marker, that is subjected to the same type of genetic modification (knockout, HDR, or base editing) as the intended edit, results in higher enrichment of successfully edited cells, than when solely selecting for transfection markers (*Katti et al., 2020*; *Coelho et al., 2018*; *Ramakrishna et al., 2014*; *Grav et al., 2015*; *St Martin et al., 2018*; *Tálas et al., 2021*; *Yan et al., 2020*) suggesting that some individual cells allow for higher editing efficiency. This phenomenon could be due to the nuclease being able to exhibit higher activity, or the activity of DNA repair systems could be considerably more engaged in these cells. We proposed that the level of enrichment, that can be achieved using the transfection marker alone, can be exceeded by enriching cells in which a marker protein in a plasmid is simultaneously prime edited. The PEAR-GFP plasmid was used in these experiments as a surrogate marker with previously optimized RT and PBS conditions (*Figure 1B*).

To test this principle, first, we transfected the BEAR-mScarlet cell line edited in *Figure 2* with the PEAR-GFP plasmid along with two pegRNAs (also coding BFP as a transfection marker) one

targeting the cell line and one targeting the plasmid to explore if prime editing on the plasmid is associated with genomic (mScarlet) editing. We measured the ratio of the mScarlet positive cells in (1) all measured cells, (2) within cells that either express BFP for transfection marker enrichment, or (3) express GFP (regardless of BFP expression) for PEAR enrichment. For this purpose, we chose two efficient PEAR-GFP targeting pegRNAs tested earlier (*Figure 1B*). *Figure 3A* shows that transfection marker enrichments significantly increased the ratio of edited and non-edited cells: gating for the BFP-expressing cells increased the percentage of the edited population from 18% to 24% with both pegRNAs tested. PEAR-enrichment resulted in a considerable increase compared to the transfection enrichments in case of both pegRNAs (57% and 76% mScarlet expressing cells, respectively). These results demonstrate that prime edited cells can be substantially enriched when using a PEAR plasmid as a surrogate marker.

To test the applicability of PEAR for enriching the prime edited population in endogenous targets generating single-nucleotide substitutions, we chose five genomic targets (EMX1, RNF2, FANCF, HEK3, and HEK4) which have been investigated previously (*Anzalone et al., 2019*). Here, BFP was used as the indicator of transfection efficiency, and we constructed a new plasmid (PEAR-GFP-2in1) as a marker for successful prime editing, which contains the pegRNA 1 (the best pegRNA in *Figure 1B*) expression cassette on the same plasmid. We co-edited the genomic targets and the PEAR-GFP-2in1 plasmid, and three days after transfection, cells were sorted into three fractions: (1) all single living cells, (2) BFP-positive cells, and (3) GFP-positive cells (*Figure 3B*). From the sorted cell populations, editing was quantified by NGS. Since HEK293T cells are very susceptible to prime editing due to the fact that their mismatch repair system is partially defective (*Chen et al., 2021*; *Trojan et al., 2002*), we also attempted enrichment in K562 and U2OS cells, and on a human embryonic stem cell line HUES9, that are all less susceptible to prime editing (*Figure 3C–E*).

The editing rates increased using transfection marker enrichment by up to 1.65-fold; however, they all showed a significantly greater increase when the PEAR enrichment was used. PEAR enrichments ranged between about twofold and fourfold, reaching up to 62% and 76% editing at FANCF site in K562 and HEK293T cells, respectively. Editing the same target in HUES9 cells showed a 7.8-fold increase with 28% PEAR-enriched editing, whilst editing without enrichment remained below 4% in HUES9 cells (*Figure 3C* and *Figure 3—source data 1*).

We also analyzed whether PEAR-enrichment alters the specificity (defined here as editing%/indel%) of editing, by quantifying the generated indels by NGS from the same samples (*Figure 3D*). Enrichments increased the incidence of indels at the edited sites in the population in 8 out of the 13 edits. However, the extent of these increments is generally low: the specificity slightly decreased for only one edit, while it slightly increased in five cases (*Figure 3E*).

Furthermore, we determined off-target prime editing from enriched samples at previously reported (*Tsai et al., 2015*; *Anzalone et al., 2019*; *Kleinstiver et al., 2016a*) SpCas9 off-target sites for four targets in HEK293T cells. Only 1 of the 15 off-target sites showed detectable editing (*Figure 3F*) and indel formation (*Figure 3G*) that was also enriched by PEAR enrichments. For the remaining 14 sites, PEAR-enrichment did not increase either off-target editing or indel formation to a detectable level. Thus, these data suggest that PEAR enrichments substantially increase prime editing in various cell lines without compromising neither on-target nor off-target specificity.

The PEAR plasmid may also contribute to undesired DNA modifications during editing. *Figure 3— figure supplement 1* shows that integration of the active plasmid to the genome is minimal; however, since plasmids tend to integrate into the genome, albeit with low efficiency, unedited PEAR plasmids are likely to still integrate into the genome. Clones should be checked for the absence of the transfected plasmids, such as the PE-expressing and the PEAR plasmid. We have developed PCR primers (*Supplementary file 1*) for detecting whether the PEAR plasmid has been integrated into the genome.

It is also important, that the pegRNA used for co-editing the PEAR plasmid has no on-target site in the genome and has only minimal genome-wide off-target effect in order to decrease the potential off-target events during PEAR-enrichments. However, to check the genome-wide off-targets of SpCas9 with such sgRNA is inherently a challenging task, since, due to the lack of a reference on-target site in the genome, it is difficult to interpret the results of a GUIDE-seq experiment aiming to determine its off-targets. To overcome this problem, we chose two EGFP targets (sites 2 and 7) that have no on-target sites in the genome but can be tested for off-targets in HEK293.EGFP cells. This cell line provides the reference on-target sites that are normally not present in the genome of

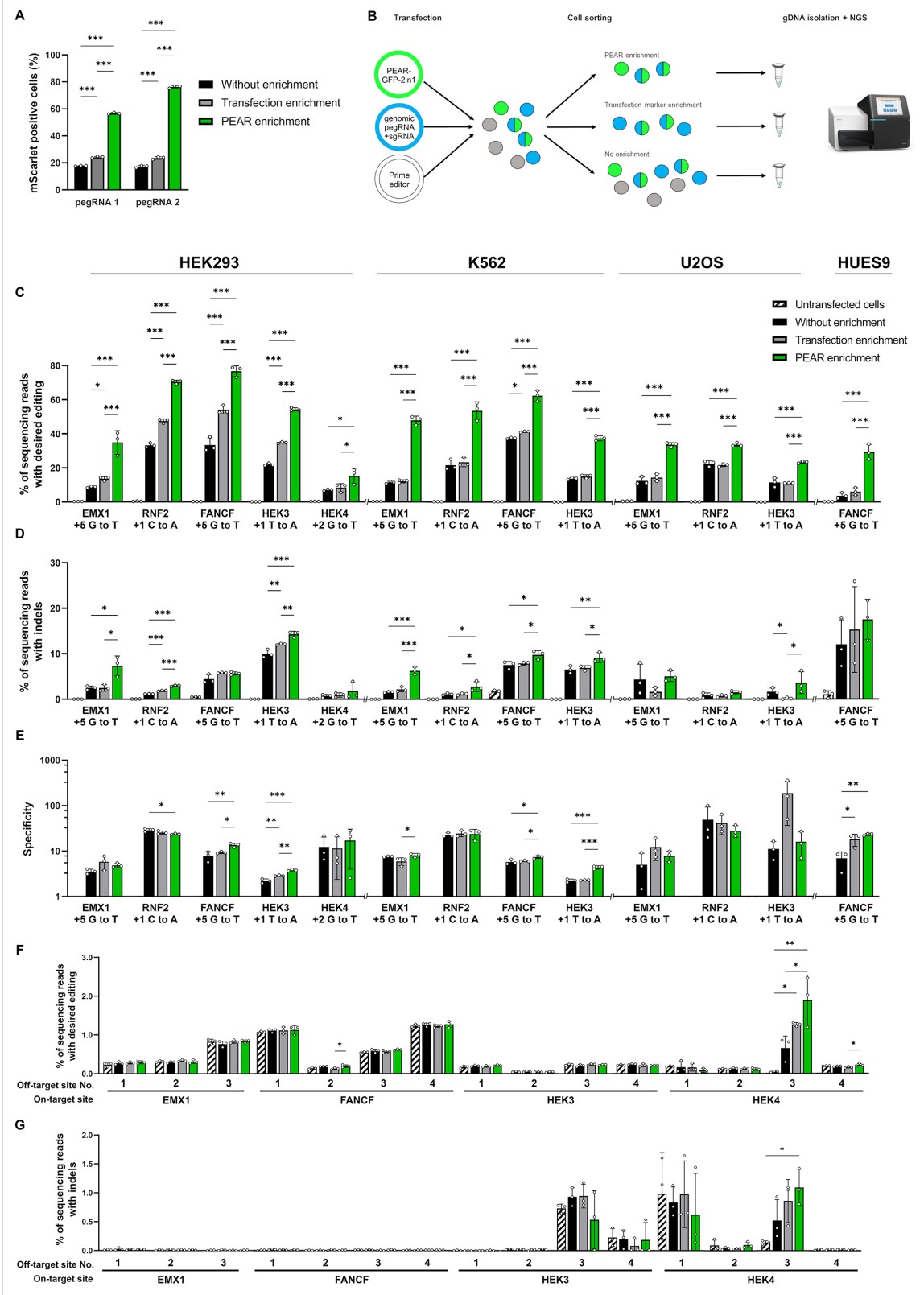

**Figure 3.** PEAR enriches prime edited cells. (**A**) Flow cytometry measurements of prime edited (i.e., mScarlet positive) cells in the HEK-BEAR-mScarlet cell line. The PEAR-GFP plasmid was co-transfected with PE protein-coding plasmid, two pegRNAs targeting either the plasmid or the genomic PEAR sequences and an sgRNA (for complementary strand nicking at position +112). To edit the PEAR-GFP plasmid, two efficient pegRNAs were selected from a previous experiment (*Figure 1B*) (pegRNA 1 and pegRNA 2 are those shown in *Figure 1B* with an RT of 24 and a PBS of 10 nucleotides, and

*Figure 3 continued on next page*

*Figure 3 continued*

an RT of 33 and a PBS of 10 nucleotides, respectively). The mScarlet positive cell count was gated for either all live single cells (no enrichment, black bars); for BFP-positive cells (transfection enrichment, gray bars); and for GFP-positive cells (PEAR enrichment, green bars). (**B**) Schematic workflow of PEAR enrichment experiments. Various cell lines were co-transfected with PEAR-GFP-2in1, a pegRNA, and a complementary nicking sgRNA targeting a genomic sequence (the latter two carrying a BFP expression cassette to monitor transfection efficiency), and the PE2-encoding plasmid. Cells expressing GFP, BFP, or both are represented as green, blue or half green/half blue circles, respectively. Cells not expressing these fluorescent proteins are shown in gray. Three days after transfection, cells were sorted into three fractions: single living cells, cells expressing BFP, and cells expressing GFP. After purification of genomic DNA and PCR amplification of the targeted amplicon, the percentage of editing was determined by NGS from each sorted sample. (**C–E**) The PEAR-GFP-2in1 plasmid and endogenous genomic targets were co-edited in HEK293T, K562, U2OS, and HUES9 cells. Cells were enriched and analyzed as described in (**B**). Results from cells without enrichment are shown in black, transfection enrichment in gray, PEAR enrichment in green, and untransfected cells in striped black and white. Precise prime editing (**C**) and unwanted indel formation (**D**) were quantified from the same samples. Specificity (prime editing%/indel%) was calculated separately for each sample (**E**). Columns represent means ± SD of three parallel transfections (white circles). When indel% was below the detection limit of NGS, specificity was calculated with 0.05% indel to avoid falsely high specificity values. Differences between samples were tested using one-way ANOVA. Only statistically significant differences are shown, differences to untransfected cells are not shown. *p<0.05, **p<0.01, ***p<0.001. (**F, G**) Off-target prime editing (**F**) and indel formation (**G**) were analyzed for known off-targets of target sites from (**C**) and (**D**) in HEK293T cells. Columns represent means ± SD of three parallel transfections (white circles). Differences between samples were tested using one-way ANOVA. Only statistically significant differences are shown. *p<0.05, **p<0.01, ***p<0.001. For all measured values, applied statistic tests and exact p values see *Figure 3—source data 1*. BEAR, base editor activity reporter; PEAR, prime editor activity reporter.

The online version of this article includes the following source data and figure supplement(s) for figure 3:

**Source data 1.** Measured values and detailed statistics for *Figure 3*.

**Figure supplement 1.** Stable integration of PEAR plasmids.

**Figure supplement 1—source data 1.** Measured values for *Figure 3—figure supplement 1*.

human cells. The genome-wide off-target sites of WT-SpCas9 nuclease for EGFP site 2 have been determined previously (*Kulcsár et al., 2020*) and for EGFP site 7, they were determined in this work by GUIDE-seq. Both sgRNAs proved to have no detectable off-targets (*Figure 4A*).

For further experiments, we constructed PEAR plasmids using these targets that can be edited without off-targets. As the GFP targets are also present in the original PEAR-GFP plasmid, by introducing silent mutations we created six mismatches in the case of EGFP site 2 and altered the PAM sequence in the case of EGFP site 7 to make sure that the pegRNA will not target the GFP coding sequence on the plasmid. To find an effective pegRNA to edit the new PEAR plasmids, three PBS lengths were tested for each spacer sequence with the 24-nucleotide long RT sequence identified in *Figure 1B* as the most efficient (*Figure 4B*). The highest editing was found with EGFP site 2 with a 12-nucleotide long PBS sequence. This plasmid—named PEAR-GFP-2in1-2.0—was used in subsequent experiments to enrich additional nine target sites installing three insertions, deletions, and substitutions each (*Figure 4C–E*). These results proved that the PEAR-GFP-2in1-2.0 plasmid is also suitable for the enrichment of prime editing and confirmed the efficiency of PEAR-enrichment for various types of editing. The editing rates by PEAR-enrichments increased from 2.1- to 4.6-fold and up to 84% (*Figure 4C*). *Figure 4D and E* show that the increase in editing rates is not accompanied with a decrease in specificity in the case of any of the nine target sites.

## Discussion

Prime editing provides a great flexibility for genomic modifications without having to supply a donor DNA or needing a double-stranded DNA break (*Anzalone et al., 2019*). This technology could revolutionize the rapidly expanding universe of genome editing. Prime editing technology has readily been translated from mammalian models (*Anzalone et al., 2019*; *Sürün et al., 2020*; *Kim et al., 2020*; *Liu et al., 2020*; *Schene et al., 2020*) to plants (*Lin et al., 2020*; *Jiang et al., 2020*; *Lu et al., 2021*; *Xu et al., 2020*) and *Drosophila* (*Bosch et al., 2020*). Despite its great potential, during the preparation of our manuscript, we have found surprisingly few studies that had demonstrated the practical use of prime editing in mammalian cells (*Sürün et al., 2020*; *Schene et al., 2020*), which is a vast underestimation of the true potential flexibility of this method. This might be due to the extensive optimization it requires and the relatively low editing efficiency that can be achieved at a subset of targets. In this aspect, any approach that can substantially increase the editing efficiency is critical to aid its generalized use. Here, we developed PEAR, a sensitive fluorescent assay for the monitoring of prime editing

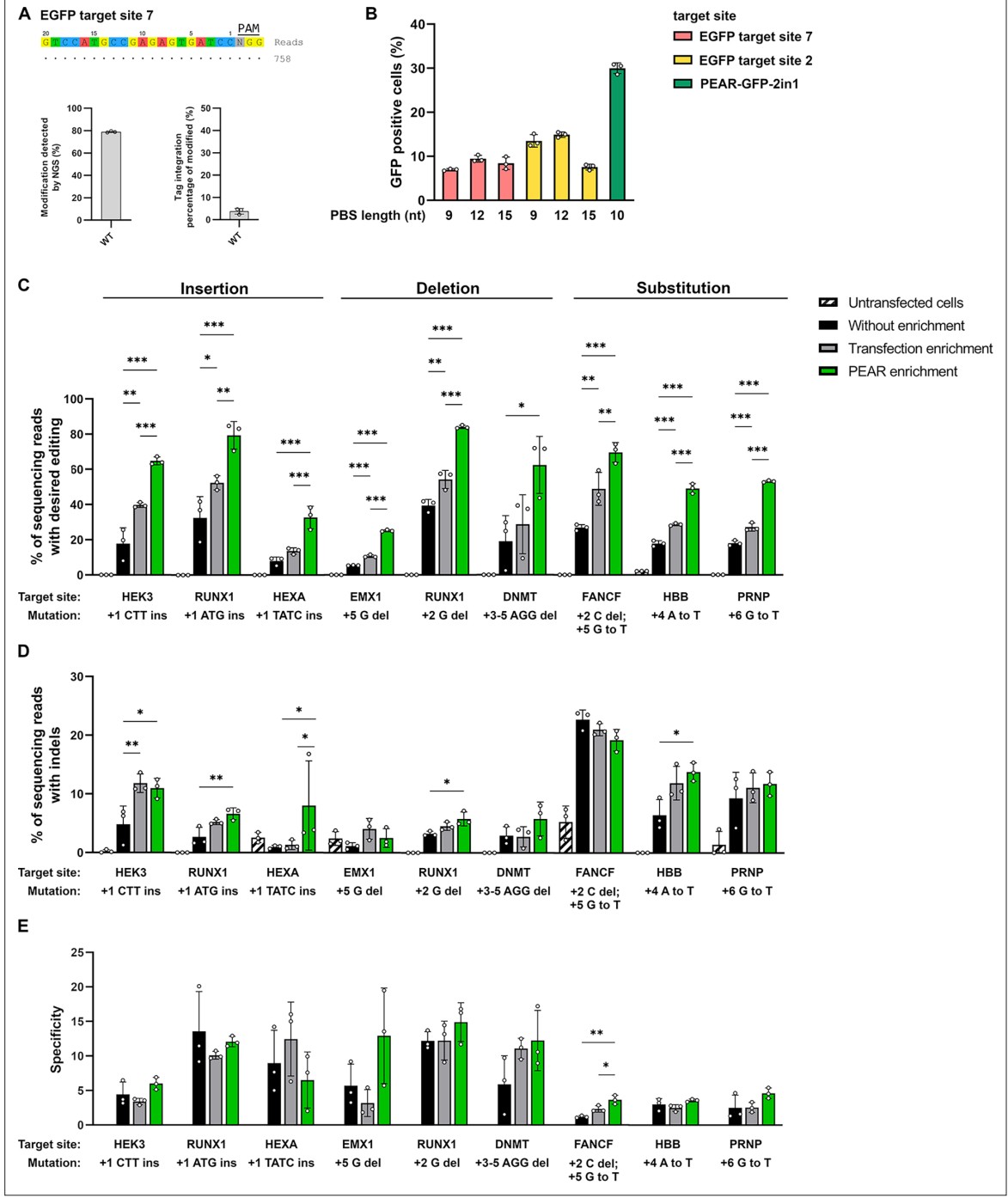

**Figure 4.** Enrichment with a PEAR reporter with no off-target activity. (**A**) Potential off-target cleavage of WT-SpCas9 was assessed by GUIDE-seq on EGFP site 7. Read counts are shown for the on-target sequence only, as no off-targets were identified. On the bar charts, the on-target genome modification (indel+tag integration) and the tag integration frequency of the modified cells are shown that were analyzed by NGS. The other analyzed site (EGFP site 2) has previously been proven to have no detectable off-target sites by GUIDE-seq (Figure 5f, **Kulcsár et al., 2020**). (**B**) Flow cytometry measurements of HEK293T cells transfected with different 2in1 PEAR plasmids, and the PE2 encoding plasmid. Target sites EGFP site 7 and EGFP site 2 (pink and yellow columns, respectively) were candidates for the construction of a PEAR-GFP-2in1 plasmid with no genome-wide off-target activity. For the plasmid targeting pegRNA, three different PBS lengths were tested with each target in combination with the 24-nucleotide long RT-template identified as the most efficient in **Figure 1B**. The PEAR-GFP-2in1 plasmid used in previous experiments is shown in green. Columns represent means ± SD of three parallel transfections (white circles). (**C–E**) PEAR can enrich all types of editing (insertions, deletions, and substitutions) in HEK293T cells. Results from cells without enrichment are shown in black, transfection enrichment in gray, PEAR enrichment in green, and untransfected cells in striped black and white. Prime editing (**C**) and indel formation (**D**) were quantified from the same samples. (**E**) Specificity (prime editing%/indel%) was calculated

*Figure 4 continued on next page*

*Figure 4 continued*

separately for each sample. Columns represent means ± SD of three parallel transfections (white circles). When indel% was below the detection limit of NGS, specificity was calculated with 0.05% indel to avoid falsely high specificity values. Differences between samples were tested using one-way ANOVA. Only statistically significant differences are shown, differences to untransfected cells are not shown. *p<0.05, **p<0.01, ***p<0.001. For all measured values, applied statistic tests and exact p values see *Figure 4—source data 1*. PEAR, prime editor activity reporter.

The online version of this article includes the following source data for figure 4:

**Source data 1.** Measured values and detailed statistics for *Figure 4*.

activity in mammalian cells, and we also showed that prime editing occurs at much higher frequencies in those individual cells, in which the PEAR plasmid is co-edited. In fact, enrichment with a transfection marker resulted in only a 1.45-fold increment on average in all experiments, whilst PEAR enrichments tripled the number of edited cells (*Figures 3A, C , and 4C*). Furthermore, we constructed a PEAR plasmid (PEAR-GFP-2in1-2.0) which bears a pegRNA that has no detectable off-target sites in the human genome as assessed by GUIDE-seq. This plasmid was used to successfully enrich additional targets with the same efficiency. Thus, the application of PEAR as a surrogate marker, as demonstrated in four cell lines including HUES9 cells, could substantially contribute to the more widespread use of prime editing by considerably increasing the number of potential targets, which may now be edited with appropriate efficiency.

PEAR is also a versatile tool that can be utilized for the characterization of different PEs. We examined more than 250 conditions combining different sgRNAs, PBS- and RT-lengths, and nick sites, and our results confirmed that the efficiency of prime editing to modify PEAR plasmids is governed by the same factors as prime editing in genomic context (*Figure 2* and *Figure 2—figure supplement 2*). The tolerance of the splice donor site for substitutions (*Figure 1D*; *Tálas et al., 2021*) makes the target region sequence of the PEAR plasmid readily adjustable. This versatility allows the user to examine the effect of sequence features and other factors affecting the efficiency of prime editing in a systematic manner that cannot be explored otherwise using genomic targets. Our approach, beyond its general use, may also assist the development of more efficient PEs or variants with enhanced characteristics in the future.

Owning to the relatively low efficiency of prime editing with many potential target sequences, PEAR is expected to become a widespread tool to be used in a wide variety of biomedical and biotechnological applications. PEAR may be an especially useful tool to be used with hard-to-edit cells or where the efficiency of PEs is particularly compromised.

## Materials and methods

### Key resources table

| Reagent type (species) or resource | Designation | Source or reference | Identifiers | Additional information |
|---|---|---|---|---|
| Cell line (*Homo sapiens*) | HEK293T | ATCC | CRL-3216 | |
| Genetic reagent (*H. sapiens*) | HEK293.EGFP | *Kulcsár et al., 2017* | | |
| Cell line (*H. sapiens*) | U2OS | ATCC | HTB-96 | |
| Cell line (*H. sapiens*) | K562 | ATCC | CCL-243 | |
| Genetic reagent (*H. sapiens*) | HEK-BEAR-GFP | *Tálas et al., 2021* | | |
| Genetic reagent (*H. sapiens*) | HEK-BEAR-mScarlet | *Tálas et al., 2021* | | |
| Cell line (*H. sapiens*) | HUES9 | Dr. Douglas Melton | | |
| Chemical compound, drug | Turbofect | Thermo Fischer Scientific Inc | R0531 | |
| Commercial assay or kit | Qubit dsDNA HS Assay Kit | Thermo Fischer Scientific Inc | Q32854 | |

*Continued on next page*

*Continued*

| Reagent type (species) or resource | Designation | Source or reference | Identifiers | Additional information |
|---|---|---|---|---|
| Commercial assay or kit | SF and SE Cell Line 4D-Nucleofector X Kit S | Lonza | V4XC-2032 V4XC-1032 | |
| Recombinant DNA reagent | Plasmids | This study | | see *Supplementary file 2* |
| Sequence-based reagents | DNA oligonucleotides | This study | | see *Supplementary file 3* |
| Chemical compound, drug | Q5 High-Fidelity DNA Polymerase | New England Biolabs Inc | M0491L | |
| Chemical compound, drug | HiFi Assembly Master Mix | New England Biolabs Inc | E2621X | |
| Chemical compound, drug | KAPA Universal qPCR Master Mix | KAPA Biosystems | KK4602 | |
| Strain, strain background (*Escherichia coli*) | NEB5-alpha competent cells | New England Biolabs Inc | C2987I | |
| Recombinant DNA reagent | pCMV-PE2 | Addgene | (#132775) | |
| Recombinant DNA reagent | pAT9624-BEAR-cloning | Addgene | (#162986) | |
| Recombinant DNA reagent | pAT9651-BEAR-GFP | Addgene | (#162989) | |
| Recombinant DNA reagent | pAT9752-BEAR-mScarlet | Addgene | (#162991) | |
| Recombinant DNA reagent | pAT9658-sgRNA-mCherry | Addgene | (#162987) | |
| Recombinant DNA reagent | pAT9679-sgRNA-BFP | Addgene | (#162988) | |
| Recombinant DNA reagent | pX330-Flag-wtSpCas9-H840A | Addgene | (#80453) | |
| Recombinant DNA reagent | pX330-Flag-dSpCas9 | Addgene | (#92113) | |
| Recombinant DNA reagent | pX330-Flag-wtSpCas9 | Addgene | (#92353) | |

## Materials

Restriction enzymes, T4 ligase, Dulbecco's modified Eagle's medium (DMEM), RPMI 1640 medium, fetal bovine serum (FBS), Turbofect, Geltrex, StemPro Accutase, Qubit dsDNA HS Assay Kit, Platinum Taq DNA polymerase, and penicillin/streptomycin were purchased from Thermo Fischer Scientific Inc SF and SE Cell Line 4D-Nucleofector X Kit S were purchased from Lonza. Bioruptor 0.5 ml microtubes for DNA Shearing were from Diagenode. Agencourt AMPure XP beads were purchased from Beckman Coulter. T4 DNA ligase (for GUIDE-seq) and end-repair mix were acquired from Enzymatics. KAPA Universal qPCR Master Mix was purchased from KAPA Biosystems. DNA oligonucleotides and the GenElute HP Plasmid Miniprep and Midiprep kit used for plasmid purifications were acquired from Sigma-Aldrich. Q5 High-Fidelity DNA Polymerase, NEB5-alpha competent cells, and HiFi Assembly Master Mix were purchased from New England Biolabs Inc.

## Plasmid construction

The PEAR-GFP plasmid (pDAS12125_PEAR-GFP) was constructed by Esp3I digestion of the pAT9624-BEAR-cloning plasmid (*Tálas et al., 2021*), followed by one-pot cloning of the linker oligonucleotides (*12125-L1 and -L2*). The reaction included 2 units of Esp3I enzyme, 1.5 units of T4 DNA ligase, 1 mM DTT, 500 µM ATP, 50 ng vector, and 5–5 µM of target-coding oligonucleotides. Components were mixed in with 1× Tango buffer, and the mixture was incubated at 37°C for 30 min before being transformed into NEB5-alpha competent cells. The active PEAR-GFP plasmid (pDAS12124_PEAR-GFP-preedited) was constructed by the above protocol using linker oligonucleotides (*12124-L1 and -L2*).

The PEAR-GFP-2in1 plasmids (pDAS12342_PEAR-GFP_2in1, pDAS12489_PEAR-GFP_2in1_2.0) and their active versions were constructed via NEB HiFi Assembly and are available from Addgene.

To monitor transfection efficiency, fluorescent protein (mCherry or TagBFP) expressing pegRNA cloning plasmids were constructed (pDAS12069-U6-pegRNA-mCherry, pDAS12222-U6-pegRNA-BFP). To construct both plasmids, an exchangeable cassette from a previously made plasmid was cloned into pAT9658 and pAT9679 between NdeI and EcoRI sites. Both pegRNA cloning plasmids bear a spacer cloning site (between BpiI sites) and a PBS-RT cloning site (between Esp3I sites).

For the construction of pegRNAs, a one-step pegRNA cloning protocol was used, described by *Anzalone et al., 2019*; however, it resulted in frequent deletions and/or mutations in the constructs,

therefore we also used alternative strategies. The most effective was a two-step cloning procedure: first, the spacer coding linkers were cloned into pDAS12069 or pDAS12222 plasmids between BpiI sites using 2 units of the BpiI enzyme, 1.5 units of T4 DNA ligase, 500 μM ATP, 1× Green buffer, 50 ng vector, and 5 μM of each annealed oligonucleotide. In the second cloning step, oligonucleotides bearing the PBS-RT were cloned into between Esp3I sites of the plasmids created in the first cloning step with the above method using 1 mM DTT and 1× Tango buffer.

For the second nicking of a plasmid or the genome, all sgRNA targets were cloned into pAT9658-sgRNA-mCherry or pAT9679-sgRNA-BFP plasmids between BpiI sites via one-pot cloning, as described above.

*Supplementary file 2* includes all plasmids *Supplementary file 3* includes all primers developed or used in this study. The sequences of all plasmid constructs were confirmed by Sanger sequencing (Microsynth AG).

Plasmids acquired from the non-profit plasmid distribution service Addgene were the following: pCMV-PE2 (#132775) (*Anzalone et al., 2019*), pAT9624-BEAR-cloning (#162986), pAT9651-BEAR-GFP (#162989), pAT9752-BEAR-mScarlet (#162991), pAT9658-sgRNA-mCherry (#162987), pAT9679-sgRNA-BFP (#162988) (*Tálas et al., 2021*), pX330-Flag-wtSpCas9-H840A (#80453), pX330-Flag-dSpCas9 (#92113), and pX330-Flag-wtSpCas9 (#92353) (*Kulcsár et al., 2017*).

The following plasmids developed in this study are available from Addgene: pDAS12125_PEAR-GFP (#177178), pDAS12124_PEAR-GFP-preedited (#177179), pDAS12069-U6-pegRNA-mCherry (#177180), pDAS12222-U6-pegRNA-BFP (#177181), pDAS12230_pegRNA-PEAR-GFP(10PBS-24RT)-mCherry (#177182), pDAS12137_sgRNA-PEAR-GFP-nick(+17)-mCherry (#177183), pDAS12342_PEAR-GFP_2in1 (#177184), pDAS12395_PEAR-GFP-2.0-preedited (#177185), and pDAS12489_PEAR-GFP_2in1-2.0 (#177186).

## Cell culturing and transfection

HEK293T (CRL-3216), U2OS (HTB-96), and K562 (CCL-243) cell lines were from ATCC. The HUES9 human embryonic stem cell line was a gift from Dr. Douglas Melton. Cell lines were authenticated by their respective suppliers and were regularly tested negative for mycoplasma.

HEK293T, HEK293.EGFP, and U2OS cells were grown in DMEM, K562 cells were grown in RPMI 1640, both media were supplemented with 10% heat-inactivated FBS with 100 units/ml penicillin and 100 μg/ml streptomycin. Cells were cultured at 37°C in a humidified atmosphere of 5% $CO_2$. HUES9 cells were maintained on Geltrex coated plates in mTeSR1 media (Stemcell Technologies) at 37°C and 5% CO2. Cultures were passaged every 4–5 days using StemPro Accutase Cell Dissociation Reagent and placed in mTeSR1 supplemented with 10 μM ROCK inhibitor (Y-27632-2HCl, Selleckchem) for the first 24 hr.

HEK293T cells were seeded on 48-well plates 1 day before transfection at a density of $5 \times 10^4$ cells/well. A total of 565 ng DNA was used: 55 ng of PEAR target plasmid, 153 ng of pegRNA-mCherry (or pegRNA-BFP in the case of BEAR-mScarlet), 49 ng of sgRNA-mCherry (or sgRNA-BFP in the case of BEAR-mScarlet), and 308 ng of PE2 coding plasmid. These were mixed with 1 μl Turbofect reagent diluted in 50 μl serum-free DMEM, and the mixture was incubated for 30 min at RT before adding it to the cells. Each transfection was performed in three replicates. Cells were analyzed by flow cytometry on day 3 from transfection.

In the enrichment experiments, where FACS was used, HEK293T cells were transfected with the above protocol. A total of 550 ng DNA was used: 40 ng of PEAR-GFP-2in1 plasmid, 153 ng of pegRNA-BFP, 49 ng of sgRNA-BFP, and 308 ng of PE2 coding plasmid. In each parallel condition a total of 12 wells were transfected. Three days after transfection, cells were trypsinized and sorted directly into genomic lysis buffer, which was followed by genomic DNA extraction.

The BEAR-GFP and BEAR-mScarlet cell lines were constructed as described earlier in *Tálas et al., 2021*. These stable cell lines were transfected with a total of 565 ng DNA: 170 ng of pegRNA-mCherry (or pegRNA-BFP in the case of BEAR-mScarlet), 54 ng of sgRNA-mCherry (or sgRNA-BFP in the case of BEAR-mScarlet), and 340 ng of PE2 coding plasmid, that was all mixed with 1 μl Turbofect reagent diluted in 50 μl serum-free DMEM. The mixture then was incubated for 30 min at RT before adding it to the cells.

In the case of U2OS and K562 cells, 1000 ng total DNA; 603 ng PE2 coding plasmid, 300 ng pegRNA-BFP, 97 ng sgRNA-BFP, and 78 ng PEAR-2in1 plasmid was nucleofected in a final volume of

20 µl in a 16-well nucleocuvette strip. 3×10⁵ U2OS and 5×10⁵ K562 cells were transfected per well using an Amaxa 4D-Nucleofector (Lonza) according to the manufacturer's protocol. For U2OS, the SE Cell Line 4D-Nucleofector X Kit (program DN-100) for K562 the SF Cell Line 4D-Nucleofector X Kit (program FF-120) was used. Three days after transfection, U2OS and K562 cells were sorted directly into genomic lysis buffer, which was followed by genomic DNA extraction.

HUES9 cells were transfected at ~70% confluence. Two hours before electroporation, the media were changed to fresh mTeSR1, then the cells were singularised by StemPro Accutase. 603 ng PE2 coding plasmid, 300 ng pegRNA-BFP, 97 ng sgRNA-BFP, and 78 ng PEAR-2in1 plasmid were nucleofected in a final volume of 20 µl in a 16-well nucleocuvette strip. 2×10⁵ cells were mixed with 20 µl homemade electroporation buffer (described in *Vriend et al., 2014*), then were electroporated with an Amaxa 4D-Nucleofector using the CA-137 program. Transfected cells were plated at Geltrex coated 48-well plates in ROCK inhibitor supplemented mTeSR1 for the first 24 hr and then the media were changed. Three days after transfection, HUES9 cells were sorted directly into genomic lysis buffer, which was followed by genomic DNA extraction.

In all experiments, the pCMV-PE2 (*Anzalone et al., 2019*) plasmid was used as the PE expressing plasmid, pX330-Flag-wtSpCas9-H840A as the nSpCas9, pX330-Flag-dSpCas9 as the dSpCas expressing plasmid, and pX330-Flag-wtSpCas9 as the WT-SpCas9 expressing plasmid. When no second nick was introduced a mock sgRNA expressing plasmid (pAT9922-mCherry or pAT9762-BFP) was used. All experiments were transfected in triplicates unless stated otherwise.

## Flow cytometry and cell sorting

Flow cytometry analysis was carried out using an Attune NxT Acoustic Focusing Cytometer (Applied Biosystems by Life Technologies). In all experiments, a minimum of 10,000 viable single cells were acquired by gating based on the side and forward light-scatter parameters. BFP, GFP, mCherry, and mScarlet signals were detected using the 405 (for BFP), 488 (for GFP), and 561 nm (for mCherry and mScarlet) diode laser for excitation, and the 440/50 (BFP), 530/30 (GFP), 620/15 (mCherry), and 585/16 nm (mScarlet) filter for emission. Attune Cytometric Software v.4.2 was used for data analysis. To compare prime editing in the cell line or on a plasmid Pearson's correlation coefficient was used.

Cell sorting was carried out on a FACSAria III cell sorter (BD Biosciences). The live single-cell fraction was acquired by gating based on side and forward light-scatter parameters. BFP or GFP signals were detected using the 405 or 488 nm diode laser for excitation and the 450/50 or 530/30 nm filter for emission, respectively. To sort control (no enrichment) cells, live single cells were sorted regardless of any fluorescent markers. To sort transfection marker enriched cells, BFP-positive cells were sorted regardless of GFP fluorescence. To sort PEAR enriched cells GFP-positive cells were sorted regardless of BFP fluorescence. A minimum of 50,000 cells were sorted in all experiments.

## GUIDE-seq

2×10⁵ HEK293.EGFP cells were resuspended in a homemade nucleofection solution (described by *Vriend et al., 2014*) and mixed with 666 ng of WT-SpCas9 expression plasmid (pX330-Flag-wtSpCas9), 334 ng of mCherry and sgRNA coding plasmid, and 30 pmol of the dsODN containing phosphorothioate bonds at both ends (according to the original GUIDE-seq protocol *Tsai et al., 2015*). Cells were nucleofected in a final volume of 20 µl in a 16-well nucleocuvette strip using the CM-130 program on a Lonza 4-D Nucleofector instrument. Transfection efficacy was analyzed by counting mCherry expressing cells after 96 hr post-transfection by flow cytometry, then genomic DNA was purified.

dsODN tag integration and efficient indel formation were verified on the on-target site by Sanger sequencing followed by TIDE (*Brinkman et al., 2014*) analysis or by NGS. The entire dsODN sequence as well as a 15-bp long centre fragment of it ('gttgtcatatgttaa'/'ttaacatatgacaac') was counted in the aligned reads to measure dsODN on-target tag integration for GUIDE-seq experiments. In the next step, genomic DNA was sheared with BioraptorPlus (Diagenode) to 550 bp in average. Sample libraries were assembled as described by *Tsai et al., 2015* and sequenced on an Illumina MiniSeq Instrument by Delta Bio 2000 Ltd. Data were analyzed using open-source GUIDE-seq software (version 1.1; *Tsai et al., 2016*). Consolidated reads were mapped to the human reference genome GRCh37 supplemented with the integrated EGFP sequence. Upon identification of the genomic regions with integrated double-stranded oligodeoxynucleotide (dsODNs) in aligned data, off-target sites were retained if at most seven mismatches against the target were present and if absent in the

background controls. Visualization of aligned off-target sites is provided as a color-coded sequence grid. GUIDE-seq sequencing data are deposited at NCBI Sequence Read Archive: PRJNA779199.

### Genomic DNA purification and genomic PCR

Genomic DNA from FACS or other experiments was extracted according to the Puregene DNA Purification protocol (Gentra Systems Inc). The purified genomic DNA was subjected to PCR analysis conducted with Q5 polymerase and locus-specific primers (see *Supplementary file 2*). PCR products were gel purified via NucleoSpin Gel and PCR Clean-up Kit (Macherey-Nagel) and were subjected to Sanger or next-generation sequencing.

### Next-generation sequencing, indel, and editing frequency analysis

In enrichment experiments, prime editing efficiency and indel frequency were analyzed by NGS (*Figures 3 and 4*). Amplicons for deep sequencing were generated using two rounds of PCR by Q5 high-fidelity polymerase to attach Illumina handles. The first step, PCR primers used to amplify target genomic sequences and indexing of samples are listed in *Supplementary file 4*. After the second step, PCR samples were quantified using Qubit dsDNA HS Assay Kit and PCR products were pooled for deep sequencing. Sequencing on an Illumina NextSeq instrument was performed by Delta Bio 2000 Ltd. Reads were aligned to the reference sequence using BBMap.

Indels were counted computationally among the aligned reads that matched at least 75% to the first 20 bp of the reference amplicon. Indels without mismatches were searched at ± 2 bp around the cut site with allowing indels of any size. For each sample, indel frequency was determined as (number of reads with an indel)/(number of total reads).

Frequency of substitution without indels generated by prime editing was determined as the percentage of (sequencing reads with the intended modification, without indels)/(number of total reads). By contrast, frequency of intended insertions or deletions generated by prime editing was determined as the percentage of (all sequencing reads with the intended modification)/(number of total reads). For these samples, the indel background was calculated from reads containing different types of indels, than the aimed edit. For NGS analysis, the following software were used: BBMap 38.08, samtools 1.8, BioPython 1.71, and PySam 0.13. To avoid falsely high specificity ratios on *Figures 3E and 4E*, during calculations indels lower than 0.05% were assumed to be 0.05% as this amount is considered to be the resolution limit of NGS. The deep sequencing data have been submitted to the NCBI Sequence Read Archive under accession number PRJNA779199.

### Statistics

Unless stated otherwise, differences between samples were tested using one-way ANOVA with Tukey's post hoc test for homoscedastic samples. Homogeneity of variances was tested by Brown-Forsythe test and normality of residuals was tested by D'Agostino-Pearson omnibus (K2) test. In cases where data did not pass normality but fulfilled the assumptions of Box-Cox transformation the transformed data were analyzed as above. If not, Kruskal-Wallis test with Dunn's test was applied. Statistical tests were performed using GraphPad Prism 9.2. Test results are shown in *Figure 3—source data 1* and in *Figure 4—source data 1*.

## Acknowledgements

The authors thank Andrew V Anzalone for providing NGS primer sequences, Vanessza Laura Végi for proofreading the manuscript and for the helpful comments on it, and Vivien Karl, Ildikó Szűcsné Pulinka, Judit Szűcs, Lilla Burkus, and Judit Kálmán for their excellent technical assistance. The project was supported by grants K128188 and K134968 to EW and PD134858 to PIK from the Hungarian Scientific Research Fund (OTKA) and DAS by KFI_16-1-2017-0232 and PIK by KFI_16-1-2017-0254 from the National Agency for Research, Development and Innovation and by ÚNKP-20-5-SE-20. PIK is a recipient of the János Bolyai Research Scholarship of the Hungarian Academy of Sciences (BO/764/20).

## Additional information

### Funding

| Funder | Grant reference number | Author |
|---|---|---|
| Hungarian Scientific Research Fund | K134968 | Ervin Welker |
| Hungarian Scientific Research Fund | K128188 | Ervin Welker |
| Hungarian Scientific Research Fund | PD134858 | Péter István Kulcsár |
| National Research, Development and Innovation Fund | 2018-1.1.1- MKI-2018-00167 | Péter István Kulcsár |
| National Research, Development and Innovation Fund | ÚNKP-20-5-SE-20 | Péter István Kulcsár |
| Hungarian Academy of Sciences | BO/764/20 | Péter István Kulcsár |

The funders had no role in study design, data collection and interpretation, or the decision to submit the work for publication.

### Author contributions

Dorottya Anna Simon, Data curation, Formal analysis, performed experiments and interpreted the results; András Tálas, Conceptualization, Data curation, Formal analysis, Writing – original draft, conceived and designed the experiments, contributed to molecular cloning, performed experiments on mammalian cells, interpreted the results and wrote the manuscript with input from all authors; Péter István Kulcsár, Data curation, Formal analysis, contributed to molecular cloning, experiments with U2OS cells and executed GUIDE-seq; Zsuzsanna Biczók, Data curation, performed experiments with stem cells; Sarah Laura Krausz, Software, analysed NGS data; György Várady, Data curation, Resources, performed cell sorting; Ervin Welker, Formal analysis, Supervision, Writing – original draft, conceived and designed the experiments, interpreted the results, supervised the research and wrote the manuscript with input from all authors

### Author ORCIDs

Dorottya Anna Simon http://orcid.org/0000-0002-0252-0072
András Tálas http://orcid.org/0000-0003-0851-8333
Ervin Welker http://orcid.org/0000-0001-9874-8794

### Decision letter and Author response

Decision letter https://doi.org/10.7554/eLife.69504.sa1
Author response https://doi.org/10.7554/eLife.69504.sa2

---

## Additional files

### Supplementary files

• Supplementary file 1. Primers and PCR condition to detect PEAR plasmid integration. Forward and reverse PCR primers, PCR product sizes, and a detailed PCR protocol suitable for detecting integrated PEAR plasmids are provided.

• Supplementary file 2. List of plasmids used in this study. This file contains all of the plasmids used in the study with references and additional comments.

• Supplementary file 3. List of oligonucleotides used in this study. This file contains the oligonucleotide sequences used for molecular cloning or PCR.

• Supplementary file 4. Sample indexing for NGS. This file contains the necessary information to identify the NGS samples used in this study.

• Transparent reporting form

## Data availability

The following plasmids developed in this study are available from Addgene: pDAS12125_PEAR-GFP (#177178), pDAS12124_PEAR-GFP-preedited (#177179), pDAS12069-U6-pegRNA-mCherry (#177180), pDAS12222-U6-pegRNA-BFP (#177181), pDAS12230_pegRNA-PEAR-GFP(10PBS-24RT)-mCherry (#177182), pDAS12137_sgRNA-PEAR-GFP-nick(+17)-mCherry (#177183). pDAS12342_PEAR-GFP_2in1 (#177184), pDAS12395_PEAR-GFP-2.0-preedited (#177185), and pDAS12489_PEAR-GFP-2in1-2.0 (#177186). GUIDE-seq and NGS sequencing data are deposited at NCBI Sequence Read Archive: PRJNA779199.

The following dataset was generated:

| Author(s) | Year | Dataset title | Dataset URL | Database and Identifier |
|---|---|---|---|---|
| Simon AD, Tálas A, Kulcsár PI, Biczók Z, Krausz SL, Várady G, Welker E | 2021 | PEAR NGS | http://www.ncbi.nlm.nih.gov/bioproject/779199 | NCBI BioProject, PRJNA779199 |

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
