## [Editor Report]

Prime editing (PE) is an emerging precision genome editing technology. It is based on the fusion of reverse transcriptase and Cas9 nickase guided by a modified guide RNA (pegRNA) to select the target site in the genomic DNA. The design of pegRNAs is not trivial and PE's editing efficiencies are low and highly dependent on the context. The authors of this manuscript have developed a surrogate reporter-based approach (PEAR) not only to identify but also to enrich for the cells that have been edited by PE. This approach will provide the means to better understand the molecular basis of PE (which could lead to the development of new, improved PEs) and improve the efficiency of PE mediated genome editing. This study, describing PEAR is of interest to the researchers in the basic biological, biomedical and agricultural sciences fields.

---

## [Decision Letter]

**Decision letter after peer review:**

Thank you for sending your article entitled "PEAR: a flexible fluorescent reporter for the identification and enrichment of successfully prime edited cells" for peer review at *eLife*. Your article is being evaluated by 4 peer reviewers, one of whom is a member of our Board of Reviewing Editors, and the evaluation is being overseen by Didier Stainier as the Senior Editor. The following individual involved in the review of your submission has agreed to reveal their identity: Jennifer Hamilton (Reviewer #3).

The main concern expressed by all four reviewers is that the PEAR technology has so far limited usefulness, at least as presented in the current version of the manuscript. In order to consider this manuscript for publication, we would like to see additional experiments demonstrating that this technology can be used to detect and enrich for PE edits in difficult-to-edit cell types and more elaborate prime edits beyond single nucleotide substitutions, such as insertions/deletions.

*Reviewer #1:*

Current reporter systems to detect prime editing outcomes are restricted to few target sequences and/or show a low signal. The aim of the study was to develop a reporter system for an easy, fluorescence-based detection of prime editing outcomes, that allows maximum flexibility for testing the target sequences and pegRNA designs in various cell types. Moreover, the authors aimed to use the developed reporter system as a plasmid-based surrogate marker to select/enrich for cells that undergo PE mediated chromosomal DNA modifications.

Strengths

The reporter is based on a split GFP protein separated by the last intron of the mouse Vim gene. The sequence of the functional 5' splice site is altered in such way, that splicing and therefore the GFP fluorescence is disrupted, however, they can be restored by prime editor harboring an appropriately designed pegRNA. In this case the DNA sequence that base pairs to the spacer region of the pegRNA is downstream of the splice site. This setup provides a gain-of-function fluorescent signal with minimal background and an unlimited flexibility to test and design pegRNAs for multiple target sequences and surrounding sequence context.

Since this setup allows to replicate the chromosomal site of interest on the surrogate reporter plasmid, it enables enrichment (~3 fold) of the cells that get edited at the chromosomal DNA site.

Since the reporter is on plasmid, it is easy to use, no need to create cell lines. In principle, it is not restricted to specific cell lines.

Weaknesses

Although the paper does have strengths, the main weakness of the paper is that not all of these strengths are directly demonstrated. In particular:

The authors imply that the surrogate reporter system they've developed is not restricted to one or a few cell lines. However, all the experiments were performed in one cell line (HEK293T).

The advantage of using prime editing instead of more efficient base editing is that this approach allows the introduction of all types of substitutions and/or precise indels. The authors demonstrated that their reporter system works well when PE is designed to substitute several nucleotides but there were no experiments performed to demonstrate how well this reporter system performs when designing pegRNAs to perform small or large indels.

1. The authors imply that the surrogate reporter system they've developed is not restricted to one or a few cell lines. The authors should demonstrate this by performing prime editing in different cells in the presence of their surrogate reporter system. It would be interesting to see if the PEAR would allow significant enrichment of PE edited cells such as K562 or U2Os which have quite low prime editing efficiencies (PMID: 31634902).

2. The authors aimed to demonstrate that indels introduced by nCas9 do not turn on GFP fluorescence (Figure 1c). However, this could result due to nCas9 being inactive during this experiment. The authors should perform an additional control experiment showing that nCas9 was active and introduced indels but at the same time did not turn on GFP fluorescence.

3. The advantage of using prime editing instead of more efficient base editing is that this approach allows the introduction of all types of substitutions and/or precise indels. The authors demonstrated that their reporter system works well when PE is designed to substitute several nucleotides. Is it possible to use this system for designing pegRNAs to perform small or large indels?

*Reviewer #2:*

Simon and colleagues report on the development of a fluorescent reporter system, PEAR, which facilitates the enrichment and isolation of cell populations edited by prime editors (PEs). The strength of this reported method is that it can increase the likelihood of identifying cells that have undergone successful prime editing at target loci. In addition, the relative ease of use of the reporter system will make it easily adoptable by other researchers. However, there are several weaknesses in the manuscript in its current form including but not limited to the following: (i) Relative limited number of target loci that were used to validate the PEAR-based system, (ii) Lack of demonstration that the PEAR-based system works across multiple cell lines including those recalcitrant to genome modification, (iii) Missing off-target analysis as it relates to PEAR-induced editing events, (iv) Insufficient statistical analysis, and (v) Missing comparisons to gold-standard methods of editing enrichment. Because of these significant weaknesses, I do not recommend the manuscript to be published in its current form.

I recommend the authors address the following points prior to publication:

(1) As it relates Figure 3b, the standard for analyzing these types of editing events would be an NGS/HTS analysis of the targeted loci to see the individual allelic outcomes from the editing. As in currently stands, the authors only perform Sanger sequencing of the bulk populations. On a related note, does PEAR-based enrichment also increase the frequency of indel formation relative to transfection or no-enrichment? The current Sanger sequencing-based analysis does not provide this sort of insight.

(2) In Figure 3b, the authors do not perform any off-target analysis of prime editor activity. At minimum, the authors should PCR amplify the most-likely off-target loci and confirm that PEAR-based enrichment does not increase likelihood of off-target events. In addition, the authors should ensure that the pegRNA and nicking sgRNA targeting the PEAR plasmids do not induce off-target edits.

(3) In Figure 3b, the authors perform comparison against reporters of transfection. However, much of the field are using reporters of expression for enriching gene editing events. The authors should make similar comparisons to demonstrate the utility of their method.

(4) The authors only perform analysis of PEAR in bulk sorted cell populations. Although such enrichment is useful the authors do not provide any analysis of clonally isolated cell populations to determine the utility of PEAR in such applications.

(5) The authors only perform analysis on 3 genomic loci which does not provide a good indication of the broad utility of PEAR-based enrichment strategies. It is recommended that the authors demonstrate the utility of PEAR-based enrichment methods in the context of additional loci including those that are recalcitrant to typical PE strategies. In addition, the authors should demonstrate the utility of PEAR in the context of other types of PE-driven edits (i.e. small deletions, insertions) in addition to single base pair changes.

(6) Is the fluorescent signal associated with the PEAR plasmids transient? Are there a certain number of passages required to lose the fluorescent signal? Along similar lines, the authors should at least discuss what strategies can be taken to ensure there is not genomic integration of the PEAR reporter plasmids.

(7) The authors only perform PEAR based enrichment with HEK293 cells, which are very amenable to gene editing. To demonstrate the broad utility of this tool the authors should perform additional experiments in other cells lines, including those such a primary cells or pluripotent stem cells which are resistant to genetic modification.

(8) Throughout the manuscript, information about statistical analysis, number of biological replicates, and other information related to scientific rigor are missing.

*Reviewer #3:*

Prime editing is an exciting new tool in the CRISPR genome editing toolbox which can introduce targeted sequence insertions, deletions and base substitutions. New strategies to enrich for prime edited cells would abrogate the need for time-consuming single-cell sorting and clonal expansion. Simon and colleagues sought to develop a method for easily recovering prime edited cells. Such a method would be useful, as prime editing thus far has only reached moderate levels of efficiency. To achieve this, the authors developed a fluorescence-on, plasmid-based reporter that, when successfully prime edited, repairs a damaged splice site and leads to expression of a fluorescent transgene.

The authors nicely demonstrate that sorting out fluorescent, reporter-positive cells enriches for successful prime editing in the cellular genome. Prime editing of the surrogate plasmid reporter robustly enriched for successful prime editing in a genomically-integrated transgene. Interestingly, this method was also successful when one pegRNA drove prime editing of the plasmid reporter and one pegRNA directed a single base change in endogenous genes (FANCF, RNF2, HEK3).

One key feature of prime editing is that it can be used to make complex genomic alterations such as templated sequence insertions. This has been used to epitope tag endogenous genes (FLAG/His6 tags) and make prime editing distinct from base editors, which aim to alter single base pairs. A weakness of the work presented by Simon and colleagues is that they limit testing of the PEAR reporter to enrichment of cells following single base pair modifications in endogenous genes. In this context, the PEAR reporter was very good at enriching for prime edited cells but it is an open question as to how well PEAR would allow for the enrichment of more complex prime edits (such as multiple base changes or sequence insertions/deletions). Additionally, it is unclear how well the PEAR reporter would enrich for prime-edited cells edited at low efficiency (<10%), where enrichment would be most valuable. Lastly, the PEAR reporter functioned well in the cell line tested (HEK293Ts) but it is unknown how this strategy would work in other immortalized cell lines or primary cells. The PEAR reporter would be a boon if it could successfully enrich for prime edited cell types that are not amenable to single cell sorting and clonal expansion (such as primary cells or cell lines that are difficult to grow up from a single cell, such as Calu-3s).

Together, the authors achieved their aim of developing a fluorescence-on reporter that allows for the enrichment of prime edited cells, both when the prime edit is made in a genomically-integrated transgene or at an endogenous genomic site. This tool will be useful for the genome editing community and/or researchers who wish to introduce minimal base pair changes in HEK293T cells.

In the discussion, the authors state that "The tolerance of the 5' splice site for substitutions makes the sequences of the target region of the PEAR plasmid easily adjustable." The manuscript would be strengthened if the authors demonstrate that this is true.

*Reviewer #4:*

The manuscript by Dorottya Simon and colleagues describes a reporter of prime editing activity, dubbed PEAR. Prime editing is a next-generation CRISPR-based genome editing strategy. Prime editing relies on a CRISPR enzyme appended with a reverse transcriptase and a stretch of single-stranded RNA that can be used as a template for a reaction that extends a genomic DNA "primer", ultimately incorporating the RNA-templated information into the genome. Some incarnations of prime editing incorporate a second, nicking CRISPR enzyme that can increase the frequency of desired outcomes. Prime editing efficiencies typically surpass those of homology-directed repair, but often lag behind those of base editing or cutting-based editing (the introduction of insertions/deletions). Prime editing outcomes remain somewhat unpredictable and the approach would greatly benefit from an improved "guidebook" that outlines the best practices for the technique. To this end, the PEAR reporter may facilitate high-throughput examination of new prime editing strategies, in turn resulting in a greater understanding of the technique.

A strength of this system is the substantial flexibility with respect to the spacer (targeting RNA / targeted DNA) sequence that can be used. A PAM (protospacer-adjacent motif) is required at the "business end" of the prime editing, but otherwise there is considerable freedom in terms of the guide RNA (spacer) that can be used. The PEAR system also allows enrichment of prime-edited populations, allowing the researcher to triple

A potential disadvantage is that the PEAR cassette – whether in plasmid form or following transfection or incorporation – will not be likely to capture all the qualities of a truly endogenous locus. Indeed, these qualities (and their diversity) play a critical role in determining why certain prime editing attempts fare drastically better than others. Loss of this diversity might limit the reporter's capacity for elucidating the determinants of efficient prime editing.

The following comments are intended to improve the manuscript.

"more precise CRISPR tools; base" … this should not be a semicolon. Consider a pair of dashes, and adjusting "developed, that can" to "developed, and these enzymes can"

"BEs'." no apostrophe needed; "of BEs" conveys the possessive.

Figure 1e: the black text can be hard to see when it is in front of the black data points.

"former observations that indels cannot, only substitution mutations" … This seems incorrect.

---

## [Author Response]

Reviewer #1:[…]1. The authors imply that the surrogate reporter system they've developed is not restricted to one or a few cell lines. The authors should demonstrate this by performing prime editing in different cells in the presence of their surrogate reporter system. It would be interesting to see if the PEAR would allow significant enrichment of PE edited cells such as K562 or U2Os which have quite low prime editing efficiencies (PMID: 31634902).

To facilitate enrichment experiments, we constructed a new PEAR-GFP plasmid which also bears its targeting pegRNA on the same plasmid. We used this plasmid (called PEAR-GFP-2in1) to perform PEAR enrichments on K562, U2OS cell lines and on a human embryonic stem cell line HUES9 on four, three and one genomic target, respectively. We have detected prime editing efficiencies and also indels from the same samples with NGS. The experiments resulted in 1.66-4.16 fold enrichment on K562, in 1.49-2.71 fold enrichment on U2OS and in 7.86 fold enrichment on the HUES9 cell lines compared to the control without enrichment. The results are discussed in the revised manuscript on pages 13-14, lines 308-334 and presented in the revised Figure 3C-E.

2. The authors aimed to demonstrate that indels introduced by nCas9 do not turn on GFP fluorescence (Figure 1c). However, this could result due to nCas9 being inactive during this experiment. The authors should perform an additional control experiment showing that nCas9 was active and introduced indels but at the same time did not turn on GFP fluorescence.

We performed an experiment in HEK293.EGFP cells in which we simultaneously targeted the BEAR-mScarlet plasmid with a pegRNA and the genome-integrated GFP sequence with sgRNA(s). This experimental design ensured that the indel-inducing activity of the different transfected SpCas9 variants could be tested simultaneously with their effect on the PEAR plasmid. The results confirmed that the PEAR plasmid reports prime editing activity exclusively and indels introduced by SpCas9 variants such as nCas9 do not turn on mScarlet fluorescence. We also examined the effect of paired nickase and WT-SpCas9 to confirm these conclusions. These results are discussed in the revised manuscript on page 8, lines 164-181 and presented in the revised Figure 1 —figure supplement 1E.

3. The advantage of using prime editing instead of more efficient base editing is that this approach allows the introduction of all types of substitutions and/or precise indels. The authors demonstrated that their reporter system works well when PE is designed to substitute several nucleotides. Is it possible to use this system for designing pegRNAs to perform small or large indels?

We designed twelve additional PEAR-GFP plasmids in which the functional splice site is disrupted such that they can be corrected by insertions (2 constructs) with different lengths and positions, by deletions (7 constructs) with different lengths and positions, or by both insertion and substitution (3 constructs). These constructs showed no fluorescence when transfected into HEK293T cells without PE2. All constructs could be corrected by prime editing with an efficiency of about 20-40%. These experiments showed that it is possible to measure insertions and deletions with the PEAR method. The results are discussed in the revised manuscript on page 8, lines 182-187 and presented in the revised Figure 1C.

Reviewer #2:I recommend the authors address the following points prior to publication:(1) As it relates Figure 3b, the standard for analyzing these types of editing events would be an NGS/HTS analysis of the targeted loci to see the individual allelic outcomes from the editing. As in currently stands, the authors only perform Sanger sequencing of the bulk populations. On a related note, does PEAR-based enrichment also increase the frequency of indel formation relative to transfection or no-enrichment? The current Sanger sequencing-based analysis does not provide this sort of insight.

In agreement with the Reviewer, in the revised paper we used NGS to detect prime editing efficiencies and also indels from the same samples. As expected, enriching for prime edited samples also increased the frequency of indel formation to a varying extent but without compromising the specificity of prime editing that we show as the ratio of clean editing / indel for all on-target editing. The corresponding results are presented on Figure 3C-G and Figure 4C-E.

(2) In Figure 3b, the authors do not perform any off-target analysis of prime editor activity. At minimum, the authors should PCR amplify the most-likely off-target loci and confirm that PEAR-based enrichment does not increase likelihood of off-target events. In addition, the authors should ensure that the pegRNA and nicking sgRNA targeting the PEAR plasmids do not induce off-target edits.

Concerning the first part of the recommendation, we analysed prime editing and indel formation on the most likely off-target loci (15 in total) of four genomic targets (HEK 3, HEK4, EMX1 and FANCF). We found no significant off-target prime editing or indel formation (compared to the untransfected control) for 14 out of 15 off-target sites. Moreover, on these sites PEAR enrichment did not increase off-target editing to a detectable level by NGS, while it increased on-target editing by 2.18-4.00 fold.

In case of HEK4 off-target site 3, we observed off-target prime editing and also indel formation. However, PEAR enrichment resulted in a 1.75 specificity ratio versus the 1.26 for the samples without enrichment, meaning that it did not compromise the specificity rather increased it to a small extent. The results are discussed in the revised manuscript on page 15, lines 335-342 and presented in the revised Figure 3F-G.

Regarding the second part of the recommendation, the main problem in evaluating the genome-wide off-targets of the PEAR-pegRNA is that it has no on-target sequence in the genome. Thus, it is not ensured that the lack of off-target editing is due to its high specificity rather than its low efficiency. We solved this problem by choosing GFP target sites which do not have a target in the human genome but could be tested for off-targets in HEK293.EGFP cells using GUIDE-seq. Both EGFP target site 2 (tested by Kulcsár et al., (2020) Figure 5f) and EGFP target site 7 (tested in this work) proved to have no detectable off-target sites. We constructed PEAR-GFP-2in1 plasmids in which the PEAR target sequence was replaced by these targets. We also introduced silent mutations in the GFP coding sequence on the plasmid to prevent the pegRNA from targeting it. Since we did not use a second nicking sgRNA on the plasmid in enrichment experiments, its off-target cleavage has never been a concern. Of the new 2in1 plasmids, the one with the highest editing activity was selected and used for enrichment of successful prime editing at 8 different genomic target sites. The results are discussed in the revised manuscript on pages 14-15, lines 350-378 and presented in the revised Figure 4.

(3) In Figure 3b, the authors perform comparison against reporters of transfection. However, much of the field are using reporters of expression for enriching gene editing events. The authors should make similar comparisons to demonstrate the utility of their method.

We are not convinced that placing the fluorescent reporters to the Cas9 expression plasmid instead of the sgRNA expression plasmid and by joining the Cas9 protein to the fluorescent protein through a 2A tag would change significantly the outcome of the transfection control experiments. Especially, since these types of experiments are affected more by other factors, such as the spectral propensity of the used fluorescent protein and the optical set up of the used FACS machine.

(4) The authors only perform analysis of PEAR in bulk sorted cell populations. Although such enrichment is useful the authors do not provide any analysis of clonally isolated cell populations to determine the utility of PEAR in such applications.

This is another comment, the implementation of which we do not consider necessary to assess the usefulness of PEAR enrichments. Although we agree with the Reviewer that analysis of clonally isolated cell populations would provide additional information, we feel that the information gain is not so substantial that we could afford to do it in the given time frame. We hope that the considerable body of experiments we have done convinces the Reviewer that the manuscript is sufficiently improved by the revision.

(5) The authors only perform analysis on 3 genomic loci which does not provide a good indication of the broad utility of PEAR-based enrichment strategies. It is recommended that the authors demonstrate the utility of PEAR-based enrichment methods in the context of additional loci including those that are recalcitrant to typical PE strategies. In addition, the authors should demonstrate the utility of PEAR in the context of other types of PE-driven edits (i.e. small deletions, insertions) in addition to single base pair changes.

In the revised manuscript, we did PEAR enrichments with 14 different prime editing modifications on 8 additional (11 in total) genomic targets in HEK293T cells. Two of these target sites (HEK site 4 and EMX site 1) are known to be recalcitrant to typical PE strategies. These modifications include substitution (7 targets), deletion (3 targets), insertion (3 targets) and insertion-deletion (1 target) mutations. These 14 prime editing modifications experiments resulted in a 2.12-4.63 fold enrichment compared to the control without enrichment. The results are discussed in the revised manuscript on pages 13-14, lines 308-328 and page 15 lines 373-378 and presented in the revised Figure 3C-E and Figure 4C-E.

(6) Is the fluorescent signal associated with the PEAR plasmids transient? Are there a certain number of passages required to lose the fluorescent signal? Along similar lines, the authors should at least discuss what strategies can be taken to ensure there is not genomic integration of the PEAR reporter plasmids.

In the timeframe of a PEAR experiment (1-3 days after transfection) the signal is expected to be mainly transient. The integration of a circular plasmid without homology arms into the genome is generally very low even at sites of Cas9 induced double stranded DNA breaks (DSB). Without DSB this is expected to be even lower. A previous study found that two weeks is sufficient for N2a cells to lose the fluorescent signal of a plasmid expressing GFP, the time it takes for plasmids and their expressing proteins to degrade completely. (Tálas et al., Sup. Figure S1).

To also address the Reviewer’s concern experimentally, GFP-positive cells from PEAR-enriched populations of 4 different genomic targets were collected by cell sorting. The sorted, 100% GFP-positive cell populations were passaged for 3 and 4 weeks (to ensure that no transient signal remained), and GFP positive cells were measured via flow cytometry. Compared to the control (untransfected cells, 0.02% GFP positive), 0.02-0.05% of cells were found in the GFP gate. These results support that the integration of active PEAR plasmids do not occur frequently, and the signal we see in the first couple of days is fully transient. However, these experiments cannot completely rule out the general, random integration of these plasmids that occurs naturally in experiments with plasmids. Therefore, we have developed PCR primers that can be used to detect plasmid integration. The primers and the PCR protocol is presented in Table 1. We are discussing this issue on page 14 lines 343-349. The results are presented in the revised Figure 3 —figure supplement 1.

(7) The authors only perform PEAR based enrichment with HEK293 cells, which are very amenable to gene editing. To demonstrate the broad utility of this tool the authors should perform additional experiments in other cells lines, including those such a primary cells or pluripotent stem cells which are resistant to genetic modification.

In response to this recommendation, we performed PEAR enrichments on K562 and U2OS cell lines and on a human embryonic stem cell line HUES9 on four, three and one genomic targets, respectively. We detected prime editing efficiencies and also indels from the same samples using NGS. The experiments resulted in 1.66-4.16 fold enrichment in K562, in 1.49-2.71 fold enrichment in U2OS and in 7.86 fold enrichment in the HUES9 cell lines compared to the controls without enrichment. The results are presented in the revised Figure 3C-E.

(8) Throughout the manuscript, information about statistical analysis, number of biological replicates, and other information related to scientific rigor are missing.

Thank you for your comment, we have added all the information on statistics and replications in parallel with the more relevant characterization of our approach.

We would like to thank to the Reviewer for their helpful comments and suggestions that helped us to improve the quality of our manuscript.

Reviewer #3:Prime editing is an exciting new tool in the CRISPR genome editing toolbox which can introduce targeted sequence insertions, deletions and base substitutions. New strategies to enrich for prime edited cells would abrogate the need for time-consuming single-cell sorting and clonal expansion. Simon and colleagues sought to develop a method for easily recovering prime edited cells. Such a method would be useful, as prime editing thus far has only reached moderate levels of efficiency. To achieve this, the authors developed a fluorescence-on, plasmid-based reporter that, when successfully prime edited, repairs a damaged splice site and leads to expression of a fluorescent transgene.The authors nicely demonstrate that sorting out fluorescent, reporter-positive cells enriches for successful prime editing in the cellular genome. Prime editing of the surrogate plasmid reporter robustly enriched for successful prime editing in a genomically-integrated transgene. Interestingly, this method was also successful when one pegRNA drove prime editing of the plasmid reporter and one pegRNA directed a single base change in endogenous genes (FANCF, RNF2, HEK3).One key feature of prime editing is that it can be used to make complex genomic alterations such as templated sequence insertions. This has been used to epitope tag endogenous genes (FLAG/His6 tags) and make prime editing distinct from base editors, which aim to alter single base pairs. A weakness of the work presented by Simon and colleagues is that they limit testing of the PEAR reporter to enrichment of cells following single base pair modifications in endogenous genes. In this context, the PEAR reporter was very good at enriching for prime edited cells but it is an open question as to how well PEAR would allow for the enrichment of more complex prime edits (such as multiple base changes or sequence insertions/deletions). Additionally, it is unclear how well the PEAR reporter would enrich for prime-edited cells edited at low efficiency (<10%), where enrichment would be most valuable.

In the revised version, we did PEAR enrichments with 14 different prime editing modifications on 8 additional (11 total) genomic targets in HEK293T cells. Two of these target sites (HEK site 4 and EMX site 1) are known to be recalcitrant to typical PE strategies. These modifications include substitution (7 targets), deletion (3 targets), insertion (3 targets) and insertion-deletion (1 target) mutations. With these 14 prime editing modifications experiments resulted in 2.12-4.63 fold enrichment compared to the control without enrichment. The results are discussed in the revised manuscript on pages 13-14, lines 308-328 and page 15 lines 373-378 and presented in the revised Figure 3C-E and Figure 4C-E.

Lastly, the PEAR reporter functioned well in the cell line tested (HEK293Ts) but it is unknown how this strategy would work in other immortalized cell lines or primary cells. The PEAR reporter would be a boon if it could successfully enrich for prime edited cell types that are not amenable to single cell sorting and clonal expansion (such as primary cells or cell lines that are difficult to grow up from a single cell, such as Calu-3s).

Unfortunately, our lab is not prepared to demonstrate the method on primary cells or Calu3. However, we performed PEAR enrichments on K562 and U2OS cell lines and on a human embryonic stem cell line, HUES9 on four, three and one genomic targets, respectively. We detected prime editing efficiencies and also indels from the same samples using NGS. The experiments resulted in 1.66-4.16 fold enrichment in K562, in 1.49-2.71 fold enrichment in U2OS and in 7.86 fold enrichment in the HUES9 cell lines compared to the controls without enrichment. The results are presented in the revised Figure 3C-E.

Together, the authors achieved their aim of developing a fluorescence-on reporter that allows for the enrichment of prime edited cells, both when the prime edit is made in a genomically-integrated transgene or at an endogenous genomic site. This tool will be useful for the genome editing community and/or researchers who wish to introduce minimal base pair changes in HEK293T cells.In the discussion, the authors state that "The tolerance of the 5' splice site for substitutions makes the sequences of the target region of the PEAR plasmid easily adjustable." The manuscript would be strengthened if the authors demonstrate that this is true.

We designed twelve additional PEAR-GFP plasmids in which the functional splice site is disrupted such that they can be corrected by insertions (2 constructs) with different lengths and positions, by deletions (7 constructs) with different lengths and positions, or by both insertion and substitution (3 constructs). These constructs showed no fluorescence when transfected into HEK293T cells without PE2. All constructs could be corrected by prime editing with an efficiency of about 20-40%. These experiments showed that it is possible to measure insertions and deletions with the PEAR method. The results are discussed in the revised manuscript on page 8, lines 182-187 and presented in the revised Figure 1C.

In addition to these experiments where the splice site itself was not modified *per se*, we have constructed 8 additional plasmids where the splice site, the 5’ flanking sequence or the 3’ flanking sequence was modified around the splice site. These inactive plasmids have shown very low to no fluorescence when transfected to HEK293T cells. Transfecting the corresponding plasmids with active splice sites (generated by molecular cloning) resulted in high percentage of GFP positive cells. When the inactive plasmids were prime edited by PE2 it resulted in 6-71% GFP positive cells indicating that prime editing can correct all constructs with varying efficiencies. The correction of the PEAR plasmid was highly dependent on the strength of the splice site itself, and the number of modified bases. The results are discussed in the revised manuscript on pages 8-9, lines 188-193 and presented in the revised Figure 1D.

We would like to thank to the Reviewer for their helpful comments and suggestions that helped us to improve the quality of our manuscript.

Reviewer #4:[…]The following comments are intended to improve the manuscript."more precise CRISPR tools; base" … this should not be a semicolon. Consider a pair of dashes, and adjusting "developed, that can" to "developed, and these enzymes can""BEs'." no apostrophe needed; "of BEs" conveys the possessive.

Thank you for catching these. In the revised manuscript it reads as:

page 3, lines 29-31: “Recently, more precise CRISPR tools – base ^89,^ and prime ^10^ editors – have been developed, and these enzymes can introduce modifications into the DNA”

page 3, line 35: “lags behind that of BEs”

Figure 1e: the black text can be hard to see when it is in front of the black data points.

In the revised manuscript we removed the flow cytometry plot from Figure 1 and placed it to Figure 1 —figure supplement 1D as we wanted to show example flow cytometry plots from each major experiment. In the revised figure all numbers are visible.

"former observations that indels cannot, only substitution mutations" … This seems incorrect.

We have removed this line and the corresponding figure from the revised manuscript and replaced it with Figure 1 —figure supplement 1E. Which we discuss on page 8, lines 164-181. The conclusion is now as follows:

“…PEAR specifically reports on prime editing activity and is insensitive to the potential presence of indel background…”

We would like to thank to the Reviewer for their helpful comments and suggestions that helped us to improve the quality of our manuscript.

References

Kulcsár, P.I., Tálas, A., Tóth, E. et al. Blackjack mutations improve the on-target activities of increased fidelity variants of SpCas9 with 5′G-extended sgRNAs. Nat Commun 11, 1223 (2020). https://doi.org/10.1038/s41467-020-15021-5

Tálas, András et al. “A convenient method to pre-screen candidate guide RNAs for CRISPR/Cas9 gene editing by NHEJ-mediated integration of a 'self-cleaving' GFP-expression plasmid.” DNA Research vol. 24,6 (2017): 609-621. doi:10.1093/dnares/dsx029